# Neuromedin S Regulates Steroidogenesis through Maintaining Mitochondrial Morphology and Function via NMUR2 in Goat Ovarian Granulosa Cells

**DOI:** 10.3390/ijms232113402

**Published:** 2022-11-02

**Authors:** Xuan Sun, Cheng Zeng, Feng Wang, Zhen Zhang, Fan Yang, Zhi-Peng Liu, Kang Li, Guo-Min Zhang

**Affiliations:** 1Jiangsu Livestock Embryo Engineering Laboratory, Nanjing Agricultural University, Nanjing 210095, China; 2College of veterinary medicine, Nanjing Agricultural University, Nanjing 210095, China

**Keywords:** NMS, granulosa cells, steroidogenesis, Hippo pathway, mitochondria

## Abstract

Neuromedin S (NMS) plays various roles in reproductive regulation, while the mechanism by which NMS regulates ovarian steroidogenesis remains unclear. In the current study, we confirmed the enhancement role of NMS in steroidogenesis in goat ovarian granulosa cells (GCs). To further explore the specific mechanism, we conducted a knockdown of NMUR2 in GCs followed by treatment with NMS and determined the effects of NMS treatment on mitochondrial morphology and function. The results found that NMS treatment increased the production of estrogen and up-regulated the expression of STAR, CYP11A1, 3BHSD, and CYP19A1, while the effects of NMS treatment were blocked by the knockdown of NMUR2 in goat GCs. Moreover, NMS treatment enhanced the fusion of mitochondria and up-regulated the expression of OPA1, MFN1, and MFN2, and increased mitochondrial membrane potential, the activity of respiratory chain enzymes and ATP production by maintaining a low expression level of mitochondrial unfolded protein response markers. The effects of NMS treatment on mitochondria were reversed by NMUR2 knockdown and NMS cotreatment. The possible mechanism of the results above was revealed by NMS treatment activating the Hippo pathway effector YAP1 and then managing the expression of phosphorylation PPARGC1A (Ser571). Together, these data showed that NMS promoted the fusion of mitochondria and protected mitochondrial function from mitochondrial unfolded protein response possibly via the NMUR2/YAP1/PPARGC1A pathway, thereby affecting the steroidogenesis of goat GCs. By elaborating the potential mechanism of NMS in regulating estrogen production in goat GCs, our results can serve as the mechanism reference for follicular growth and development.

## 1. Introduction

The mammalian ovary is a dynamic reproductive endocrine organ responsible for producing the ovum and providing sex steroids required for female fertility and quality of life. The growth and development of ovarian follicles require a series of coordinated events that induce morphological and functional changes within the follicle [1]. Since the endocrine mechanisms of the hypothalamic–pituitary–ovarian axis were elucidated, granulosa cells (GCs) have been the focus of interest in numerous studies that examined the mechanisms of follicular growth. GCs are somatic components of the follicle, and they make physical contact with oocytes to support their growth, maturation, and ultimately ovulation. In addition, GCs respond to stimuli from the pituitary and neighboring cells by promoting follicle growth, the secretion of sex hormones, and growth factors that are vital for reproduction [2]. Moreover, intraovarian steroid hormones can regulate follicular development by improving the follicular microenvironment. Although rapid progress utilized by the classical physiological model systems (mice and rat) has been made in regulating ovarian follicle development, the exact mechanisms underlying the regulation of steroid hormone and follicle development are not fully understood.

Neuromedin S (NMS), a novel neuropeptide of 36 amino acids, is an endogenous ligand for the orphan G-protein-coupled receptors (GPCRs) neuromedin U receptor type-1 (NMUR1) and type-2 (NMUR2), which participates in many physiological functions of the organism [3]. Soon after identifying NMS and its receptor, its distribution was described in various species via immunohistochemical methods, in situ hybridization, and autoradiography. The obtained results have consistently reported that NMS expression is mainly observed in the brain, NMUR1 mainly presents in the peripheral nervous system, and NMUR2 mainly occurs in the central nervous system [4,5,6,7]. However, evidence revealed that NMS and NMUR2 mRNAs were expressed in the pig’s hypothalamic–pituitary–ovarian axis, and noticed that NMS could regulate the reproductive axis hormone secretions of female pigs [8,9], suggesting that NMS and its receptor might have a role in the regulation of the reproductive axis to impact follicular development, but its regulatory mechanism still remains unclear.

Previous studies have found that steroidogenic acute regulatory protein (STAR) and cytochrome P450, family 11, subfamily A, polypeptide 1(CYP11A1) are located on the mitochondria, which orchestrate steroidogenesis in GCs and play an essential role in the growth and development of follicles [10]. Recent studies found that the Hippo pathway, which consists of mammalian sterile 20-like 1 (MST1), mammalian sterile 20-like 2 (MST2), large tumor suppressor kinase 1 (LATS1), large tumor suppressor kinase 2 (LATS2), and Yes1 associated transcriptional regulator (YAP1) was involved in the regulation of cellular functions via activation by GPCRs [11,12]. Moreover, a series of studies demonstrated that regulation of the Hippo pathway by GPCRs is a universal response of cells to hormonal cues [13,14,15], and as a homologue of NMS, neuromedin U (NMU) can perform physiological functions through the Hippo pathway [16]. Furthermore, studies proved that the Hippo pathway effector YAP1 was required for cell proliferation and secretion of androgen and estrogen in mouse GCs [17] and controlled the expression of PPARG coactivator 1 alpha (PPARGC1A) to regulate mitochondrial biogenesis, dynamics and functions [18]. In addition, PPARGC1A participated in mitochondrial biogenesis and the steroidogenesis of ovarian GCs [19,20]. Although numerous studies are mentioned above, there is no clear evidence that the GPCRs/Hippo pathway cooperates with PPARGC1A to affect steroidogenesis via regulating mitochondrial morphology and function in goat GCs.

In our study, we first cloned and analyzed NMS and its receptor genes and investigated the role of NMS and its receptor in goat ovarian GCs. Loss-of-function and NMS treatment experiments revealed that NMS activated NMUR2 to maintain mitochondrial morphology and function and then regulate estrogen production. In addition, we determined the effects of NMS treatment on the Hippo pathway, thereby elucidating the possible mechanism underlying the effect of estrogen production under NMS in goat ovarian GCs. By utilizing the small ruminant model system (goat), our results uncovered that NMS regulates steroidogenesis by maintaining mitochondrial morphology and function, possibly through the NMUR2/YAP1/PPARGC1A pathway, further serving as the mechanism reference for human follicular growth and development.

## 2. Results

### 2.1. Gene Cloning and Sequence Analysis

To study the functions of NMS in goat ovaries, we first cloned NMS, NMUR1, and NMUR2 gene fragments that included gene span coding sequences (CDS) regions, and obtained products with sizes of 577 bp, 1383 bp, and 1270 bp, respectively (Figure 1A). The amplified sequences of the CDS region shared more than 98.9% identities with the NCBI reference sequences (Appendix A). The phylogenetic analyses suggested that NMS and its receptor nucleotide sequences mainly remain conserved in other species and indicated high homology with *Ovis aries*, *Bos taurus*, and *Sus scrofa*, while less sequence identity with *Mus musculus* and *Gallus gallus* (Figure 1B). The transmembrane domain features indicated that goat NMUR1 and NMUR2 are seven-transmembrane receptors (Figure 1C). In addition, multiple alignments of the protein sequences demonstrated that goat NMS is relatively conserved in other vertebrates. The transmembrane amino acid composition of NMS receptors presented high levels of homology compared with other vertebrates (Appendix A). These results indicate that NMS and its receptor could play a similarly important role in other species as in goats.

### 2.2. NMS and Its Receptor Are Expressed during Goat Follicle Development

To understand the possible functional role of NMS in goat ovaries, we analyzed the mRNA and protein expression of NMS and its receptor in the ovaries of three-month-old and nine-month-old goats. As shown in Figure 2A,B, the mRNA and protein expression of NMS and NMUR2 were significantly higher in the ovaries of nine-month-old goats than in the ovaries of three-month-old goats (*p* < 0.05). In contrast, the NMUR1 expression showed no significant difference (*p* > 0.05). The IHC analysis observed that NMS and its receptor were expressed in follicles at different developmental stages, and their signals were highly stained in GCs of antral follicles (Figure 2C). Subsequently, the IF analysis results showed that NMS was mainly localized in the cytoplasm, and its receptor was expressed in both the nucleus and cytoplasm of goat GCs (Figure 2D). In addition, we analyzed the mRNA and protein levels of NMS and its receptor in follicles of different diameters in nine-month-old goat ovaries. As shown in Figure 2E,F, the mRNA and protein levels of NMS and NMUR2 were significantly increased with an increase in follicle diameter (*p* < 0.05), and the protein expression level of NMUR1 significantly decreased with an increase in follicle diameter (*p* < 0.05). These findings indicate that NMS might be an essential factor in goat ovarian follicle development.

### 2.3. NMS Promotes Steroidogenesis via NMUR2 in Goat Ovarian Granulosa Cells

To investigate the effects of NMS on steroidogenesis in goat GCs, GCs were first exposed to NMS (10^−12^, 10^−10^, 10^−8^ and 10^−6^ M) for 48 h and the concentration of estrogen in the cell supernatant was measured. It showed that the production of estrogen was significantly increased with exposure at 10^−12^, 10^−10^ and 10^−8^ M NMS (*p* < 0.05, Figure 3A), and the enhancement effect of estrogen was the most obvious under the 10^−12^ M NMS treatment. We also observed that the mRNA and protein expression of key steroid synthesis genes (STAR, CYP11A1, 3BHSD, and CYP19A1) were significantly up-regulated with exposure at 10^−12^, 10^−10^ and 10^−8^ M NMS (*p* < 0.05, Figure 3B,C), except for the mRNA level of CYP11A1 under the 10^−8^ M NMS treatment. Subsequently, GCs were exposed to 10^−12^ M NMS for 3, 6, 12, 24 and 48 h, and the result revealed that NMS increased the production of estrogen in a time-dependent manner from 3 h to 24 h (*p* < 0.05, Figure 3D), and there was no significant difference for 48 h (*p* > 0.05). A similar trend in the mRNA and protein expression of key steroid synthesis genes was observed in a time-dependent manner with estrogen production (*p* < 0.05, Figure 3E,F), except for the mRNA level of CYP11A1 and 3BHSD under the 48 h treatment. These findings suggest that NMS can stimulate estrogen synthesis, and its optimal concentration and stimulation time were 10^−12^ M and 24 h in goat GCs, respectively. Hence, the GCs were treated with 10^−12^ M NMS for 24 h in subsequent experiments.

To determine the binding receptor of NMS to promote steroidogenesis in goat GCs, we screened and transfected siRNA into GCs to knockdown NMUR1 and NMUR2 (Appendix A), then checked the production of estrogen and the mRNA and protein expressions of key steroid synthesis genes. Compared to the control, the production of estrogen and the mRNA and protein expressions of key steroid synthesis genes showed no difference in GCs with siNMUR1 treatment (*p* > 0.05, Figure 3G–I). Compared with the control group, the production of estrogen was significantly decreased (*p* < 0.05, Figure 3J), and the mRNA and protein expression of key steroid synthesis genes were significantly down-regulated in GCs with siNMUR2 treatment (*p* < 0.05, Figure 3K,L). Furthermore, NMS treatment had no effect on the production of estrogen and the protein expression of key steroid synthesis genes in GCs under siNMUR2 treatment (*p* > 0.05, Figure 3M,N). These results indicate that NMS mainly promotes estrogen synthesis by binding to NMUR2 in goat GCs.

### 2.4. NMS Regulates Mitochondrial Morphology to Affect Steroidogenesis by Mitochondrial Dynamics in Goat Ovarian Granulosa Cells

To explore the effect of NMS on mitochondrial morphology in goat GCs, we first visualized the mitochondrial morphology using the mitochondrial probe CMX-ROS under confocal microscopy. As shown in Figure 4A, NMS treatment exhibited evident hyper-fused mitochondria, while more fractions of fragmented mitochondria in poorly connected networks were observed in GCs under siNMUR2 treatment. The scoring of fusion and fission mitochondria indicated that cells opted for mitochondrial fusion in response to NMS treatment and mitochondrial fission in response to siNMUR2 treatment (Figure 4B,C). Simultaneously, we found that NMS treatment significantly increased mitochondrial area, perimeter, aspect ratio, and form factor, while these effects were blocked by siNMUR2 treatment, respectively (*p* < 0.05, Figure 4D–G). Electron microscopy confirmed the morphology changes through a fluorescence probe (Figure 4H). In basal conditions, mitochondria had an orthodox structure with narrow cristae and a more spherical shape. The cell’s mitochondria in NMS were more significant in diameter and appeared elongated and tubular as compared to the round mitochondria found in the control cells (Figure 4H). The cell’s mitochondria with siNMUR2 treatment increased damaged to mitochondria with diffuse cristae, or a pale matrix, indicating mitochondrial network fragmentation (Figure 4H). Quantitative analyses showed that the cristae densities of the cells with NMS treatment were significantly higher than the cells with siNMUR2 and NMS cotreatment (*p* < 0.05, Figure 4I). In addition, we found that NMS treatment significantly promoted the mRNA and protein expressions of the mitochondrial fusion genes MFN1, MFN2, and OPA1, while it significantly inhibited the mRNA and protein expressions of the mitochondrial fission genes DNM1L and FIS1 (*p* < 0.05, Figure 4J,K). In addition, the effect of NMS treatment on the mRNA and protein expressions of mitochondrial fusion and fission genes were reversed by NMS and siNMUR2 cotreatment (*p* < 0.05, Figure 4J,K). Together, these results demonstrated that NMS might manage steroid hormone synthesis by regulating mitochondrial dynamics and mitochondrial morphology in goat GCs.

### 2.5. NMS Protects Mitochondrial Function via Mitochondrial Unfolded Protein Response in Goat Ovarian Granulosa Cells

To explore the effect of NMS on mitochondrial function in goat GCs, the MMP, the activities of MRC complexes, ATP, and cellular ROS were detected. The results exhibited that NMS treatment significantly increased mitochondrial membrane potential, the activities of the MRC complexes CS, CI, CII, CIII, CIV, and the production of ATP, while decreasing the production of ROS (*p* < 0.05, Figure 5A–D). In addition, the effects of NMS treatment on the MMP, the activities of MRC complexes, ATP, and cellular ROS were blocked by NMS and siNMUR2 cotreatment (*p* < 0.05, Figure 5A–D). Furthermore, we observed that NMS treatment maintained a low expression level of mitochondrial unfolded protein response markers of ClpP, PKR, phospho-JNK (Thr183 + Tyr185), phospho-JUN (Ser73), and HSP60, while the expression level of mitochondrial unfolded protein response markers was restored by siNMUR2 and NMS cotreatment (*p* < 0.05, Figure 5E,F). These findings suggest that NMS could maintain mitochondrial function to affect steroid hormone synthesis through mitochondrial unfolded protein response in goat GCs.

### 2.6. NMS Might Regulate Steroidogenesis through NMUR2/YAP1/PPARGC1A Pathway in Goat Ovarian Granulosa Cells

To investigate the possible molecular mechanisms underlying the estrogen secreted by the NMS/NMUR2 system in goat GCs, we first examined the role of the Hippo pathway in ovarian follicle development. As shown in Figure 6A,B, compared to the ovaries of three-month-old goats, the mRNA and protein expressions of MST1, MST2 and LATS1 were significantly decreased (*p* < 0.05). In contrast, the mRNA and protein expressions of YAP1 significantly increased (*p* < 0.05) in the ovaries of nine-month-old goats. The IHC analysis showed that the core components of the Hippo pathway were expressed in follicles at different developmental stages, and MST2, LATS2, and YAP1 signals were highly stained in GCs of antral follicles (Figure 6C). The IF analysis observed that MST1, MST2, LATS1, and LATS2 were mainly localized in the cytoplasm, and YAP1 was mainly expressed in the nucleus in goat GCs (Figure 6D). Additionally, we analyzed the mRNA and protein levels of the core components of the Hippo pathway in follicles of different diameters in nine-month-old goat ovaries. As shown in Figure 6E,F, the mRNA and protein levels of MST1, MST2, LATS1, and LATS2 were significantly decreased with an increase in follicle diameter. In contrast, the expression of YAP1 had an opposite expression trend (*p* < 0.05). These results indicate that the Hippo pathway might play an essential role in goat ovarian follicle development.

To identify whether the Hippo pathway is a crucial pathway triggered by the NMS/NMUR2 system to regulate the secretion of estrogen, we further analyzed the protein expression of the core Hippo pathway components in GCs under NMS with or without siNMUR2 treatment. As shown in Figure 7A–C, compared with the control group, NMS treatment significantly reduced the expression level of MST1, MST2, hosphor-MST1/2 (Thr183), LATS1, LATS2, hosphor-LATS1/2 (Ser909/872), and hosphor-YAP1 (Ser127), while it significantly enhanced the expression level of YAP1 (*p* < 0.05, respectively). In addition, the effect of NMS treatment on the protein expression of the Hippo pathway core components was reversed by NMS with siNMUR2 treatment (*p* < 0.05, Figure 7A–C). Subsequently, we performed the gain-of-function and loss-of-function experiments on YAP1 in goat GCs (Appendix A). We examined the production of estrogen and the protein expression of key steroid synthesis genes. The knockdown of YAP1 significantly decreased the production of estrogen and down-regulated the protein expression of key steroid synthesis genes compared to the control (*p* < 0.05, Figure 7D,E), which was consistent with the result of siNMUR2 treatment (Figure 3M,N). The overexpression of YAP1 removed the suppressive effects of siNMUR2 treatment and restored the production of estrogen and the protein expression of STAR, 3BHSD, and CYP19A1 in goat GCs (*p* < 0.05, Figure 7F,G). Moreover, we observed that the YAP1 protein interacted with the PPARGC1A protein (Figure 7H). NMS treatment significantly reduced the ratio of phospho-PPARGC1A/PPARGC1A (phosphorylation at Ser571 was inhibited the function of PPARGC1A), while the ratio of phospho-PPARGC1A/PPARGC1A was significantly enhanced with the knockdown of NMUR2 or YAP1, respectively (*p* < 0.05, Figure 7I). Collectively, these data indicated that NMS might regulate estrogen synthesis via the NMUR2/YAP1/PPARGC1A pathway in goat GCs.

## 3. Discussion

As a neuroregulatory substance in mammals, NMS is involved in regulating a variety of physiological functions. Our study identified that NMS remained broadly conserved among species and was highly expressed during goat ovarian follicle development. Furthermore, NMS promoted mitochondrial biogenesis, induced the fusion of mitochondria, and protected mitochondrial function from mitochondrial unfolded protein response, possibly via the NMUR2/YAP1/PPARGC1A pathway, thereby affecting estrogen synthesis to regulate follicular development (Figure 8). All these findings could serve as the mechanism reference for human follicular growth and development.

Our study showed that NMS and its receptor nucleotide sequences mainly remained broadly conserved among species, and NMUR1 and NMUR2 had the seven-transmembrane domains, which is consistent with previous reports [21,22,23]. NMS regulated the release of LH and FSH from the anterior pituitary [24], and NMS and its receptor were expressed in ovaries [8], suggesting that NMS and its receptor are involved in reproductive regulation, which was supported by our results that NMS and NMUR2 were highly expressed with goat ovarian follicle development. In addition, our study found that a certain amount of NMS (mainly 10^−12^ M) promoted the secretion of estrogen from goat GCs and up-regulated the expression of STAR, CYP11A1, 3BHSD, and CYP19A1, which is consistent with the previous studies showing that NMS increased the release of testosterone [24] and corticosterone [25] from steroid-hormone-producing cells. Moreover, NMS is an isoform of NMU and plays many physiological roles similar to NMU by binding to NMUR1 and NMUR2. As reported, NMU can act directly on β cells through NMUR1 to affect insulin secretion [26,27], and both NMU and NMS can regulate the synthesis and release of LH and FSH from the pituitary through NMUR2 [24,28,29]. These results suggest that NMS may regulate different types of hormones through different receptors. Our study supported that NMS regulated steroidogenesis in goat GCs through NMUR2 rather than through NMUR1.

Mitochondrial plays a critical role in regulating steroid hormones in ovarian GCs. Our results found that NMS combined with NMUR2 maintained mitochondrial dynamics by promoting the expression of proteins related to mitochondrial biogenesis and fusion. Mitochondrial dynamics were the key to its quality control, which depends on balancing the continuous internal fusion and division process to maintain the morphology and function of mitochondria [30]. In mammalian cells, MFN1, MFN2, and OPA1 mediate mitochondrial membrane fusion, and DNM1L and FIS1, as its outer membrane adaptors, mainly anchor triggered mitochondrial fission [31]. In our study, NMS enhanced the expression of MFN1, MFN2, and OPA1, while reducing the expression of DNM1L and FIS1, causing the mitochondrial volume to become larger, thereby affecting steroidogenesis. Consistent with previous studies, steroid hormone production has been reported to be accompanied by increased mitochondrial mass, specifically an increase in mitochondrial fusion, whereas a reduction occurs in mitochondrial fission [32,33].

Moreover, the dynamic regulation of mitochondrial morphology involves regulated processes of fusion and fission, which modifies mitochondrial function [34]. A previous study found that the down-regulation of OPA1 or the invalidation of MFN1 and MFN2 strongly decreased mitochondrial respiratory capacity in cells when fueled with substrate for electron transport chain complexes I, II, and IV [35,36]. These results were also supported by our study showing that NMS enhanced mitochondrial membrane potential, the activity of respiratory chain enzymes, and ATP production, suggesting that the change in mitochondrial morphology was closely related to the functionality of mitochondria. Additionally, the knockdown of NMUR2 enhanced cellular ROS production and resulted in mitochondrial dysfunction, and the reason for this might be mitochondrial fragmentation [37]. Given that mitochondrial dysfunction and changes in morphology can trigger the mitochondrial unfolded protein response to relieve/resolve the stress and to compensate for the defect. Our results further showed that the mitochondrial unfolded protein response responded under NMUR2 knockdown and NMS cotreatment, suggesting that NMS might act as a therapeutic agent for mitochondrial stress.

The Hippo pathway plays an instrumental role in regulating follicular growth and development. In our study, the expression of MST1/2 and LATS1/2 decreased, and YAP1 expression increased with an increase in follicle diameter, suggesting that the development of primordial follicles was accompanied by an inhibition of the Hippo pathway, which is consistent with previous studies [38,39]. Moreover, multiple studies have identified a wide range of upstream regulators of the Hippo pathway, including mechanical cues and ligands of GPCRs [13,14,15]. Furthermore, many neuroregulatory substances have been demonstrated to be related to the Hippo pathway and involved in various physiological functions. The Hippo pathway participates in the luteinizing hormone [40] and gonadotropin-releasing hormone [41] secretion in gonadotrope cells. The neuropeptide Galanin contributes to cholangiocyte proliferation through activation of the Hippo pathway [42]. The expression of enkephalin is regulated by the Hippo pathway in high-fat diets during pregnancy [43]. These studies suggested that NMS as a neuroregulatory substance and NMUR2 as a CPCR performed various functions, possibly by activating the Hippo pathway [12,44,45], which was supported by our results showing that the expression of the Hippo pathway core components had significant differences under NMS with or without siNMUR2 treatment. Notably, as the crucial effector of the Hippo pathway, YAP1 played an essential role in regulating cell secretion function. As reported, the knockdown of YAP1 effectively reduced the FSH-stimulated production of estrogen and aromatase in bovine and human GCs, which was consistent with our results. Additionally, G-protein-coupled estrogen receptor could regulate follicular development by interplaying between the PI3K/AKT/mTOR axis and the Hippo pathway [41,46]. Nevertheless, whether the NMS/NMUR2 system can improve or stabilize estrogen synthesis and regulate follicular development through YAP1 converging with other pathways remains to be further studied.

YAP1 can play an essential role in cell metabolism through cooperation with PPARGC1A [18,47], which was supported by our results showing that the YAP1 protein interacted with the PPARGC1A protein in goat GCs. Moreover, PPARGC1A is central to the mitochondrial regulatory network, which performs various cellular functions. Both the expression and activity of PPARGC1A are precisely modulated to maintain its temporal and tissue-specific functions in response to diverse environmental demands [48]. Studies have proposed that the activity and stability of PPARGC1A are subject to posttranslational phosphorylation modifications, and the phosphorylation of PPARGC1A at Ser571 can decrease PPARGC1A activity [49]. Hence, our results suggested that NMS enhanced the function of PPARGC1A by directly inhibiting phosphorylation at Ser571. In addition, PPARGC1A also plays a vital role in ovarian folliculogenesis and steroidogenesis. Higher levels of expression of PPARGC1A were detected in GCs of large follicles compared with early follicular growth, and the overexpression of PPARGC1A increased steroidogenesis-related gene expression and progesterone production in KGN cells [50].

Steroidogenic enzymes orchestrate estrogen biosynthesis from cholesterol in ovarian GCs. Cholesterol is transported from the outer to the inner mitochondrial membrane by STAR, an initial and rate-limiting step. It is converted to pregnenolone by CYP11A1 located in the matrix side of the inner mitochondrial membrane. Then, pregnenolone is metabolized into estrogen by 3BHSD and CYP19A1 in the smooth endoplasmic reticulum. Mitochondria are a crucial control point for regulating steroid hormone biosynthesis, and their morphology is critically vital for maintaining steroidogenic enzyme function. As reported, MFN2 knockdown and mitochondrial fusion inhibition diminished STAR, CYP11A1 mRNA levels, and concomitantly the mitochondrial STAR protein in Leydig cells [51,52], which is supported by our study that NMUR2 knockdown decreased the expression of MFN2, STAR, and CYP11A1. In addition, the knockdown of ClpP and HSP60 could change the mitochondrial dynamics through the increase in mitochondrial fission and result in mitochondrial dysfunction [53,54]. CYP11A1 was involved in mitochondrial cristae remodeling though interaction with HSP60 [55]. Furthermore, both ClpP and HSP60 were responsible for STAR turnover in mitochondria to participate in steroidogenesis [54,56]. The significant expression changes of ClpP, HSP60, STAR, CYP11A1, mitochondrial cristae, mitochondrial morphology, and function were observed in our study, suggesting that mitochondrial unfolded protein response is related to the regulation of steroidogenesis. Hence, we propose that the maintenance of mitochondrial morphology and function via mitochondrial unfolded protein response is required to regulate steroidogenesis.

## 4. Material and Methods

All experiments were conducted according to approved Guidelines for Animal Experiments of Nanjing Agricultural University and were approved by the Animal Care and Use Committee of Nanjing Agricultural University (Approval ID: SYXK2011-0036). All antibodies were purchased from commercial suppliers, and other chemicals were obtained commercially and of reagent grade. The NMS (C133YEL260-1) of goats was synthesized by GenScript ProBio (Singapore) Ltd., Shanghai, China. The purity of NMS was more than 95% as analyzed by high-performance liquid chromatography.

### 4.1. Goat Ovaries and Follicles Collection

Goats were supplied by Haimen Goat Industry Co., Ltd., Nanjing, China, and we selected eighteen female goats assigned to one of two groups (age, 3 ± 0.1 months vs. 9 ± 0.1 months, nine goats per age group, respectively). All goats were subjected to ovariectomy to obtain ovaries as described previously [57]. A portion of ovaries in each group was fixed with Bouin’s fixative solution for immunohistochemistry (IHC) assay, and the remaining tissue was immediately frozen in liquid nitrogen for subsequent mRNA and protein expression analysis. Follicles were isolated from the 9-month-old goat ovaries and selected healthy parts to be classified into three sizes (≤2, 2–5 and ≥5 mm, at least 80, 60, and 40 per size, respectively) as described previously [57]. Separated follicles were snap-frozen in liquid nitrogen for further analysis of gene and protein expression analysis.

### 4.2. Gene Cloning and Sequence Analysis

TRIzol™ reagent was used to extract total RNA (15596018, Invitrogen, CA, USA) and RNA concentration was determined by using the NANODROP 2000 spectrophotometer (Thermo Scientific, Waltham, MA, USA). cDNA was synthesized by the PrimeScript™ RT reagent Kit with gDNA Eraser (RR047A, TaKaRa Biotechnology, Dalian, China). All cDNA samples were stored at −20 degrees Celsius.

The cloning method was conducted based on former research [58]. The specific sequence primers are listed in Appendix A. Positive clones were randomly chosen and commercially sequenced through General Biological Technology (Beijing Tsingke Biotechnology Ltd., Beijing, China). The obtained sequences of NMS, NMUR1 and NMUR2 were uploaded and aligned with NCBI reference sequences using BLAST (http://www.ncbi.nlm.nih.gov/blast, accessed on 13 September 2022). Phylogenetic tree homology assessments were performed using the neighbor-joining and dislocation comparison method in MEGA-X software. Amino acid sequences were predicted with the Expasy Proteomics Server (http://www.expasy.ch/tools/dna.html, accessed on 28 February 2021) and aligned with other species using GenomeNet (http://www.genome.jp/tools/clustalw, accessed on 9 May 2021). The accession numbers of cloning genes for five species were listed in Appendix A. Transmembrane domains were predicted by using TMHMM (http://www.cbs.dtu.dk/services/TMHMM-2.0, accessed on 9 May 2021).

### 4.3. Cell Culture and Treatments

GCs were isolated and cultured according to our previously described method [59]. In short, goat ovaries were harvested from the abattoir, and GCs were isolated from healthy follicles (2–5 mm) by a 24-gauge needle attached to a 10 mL syringe. GCs were cultured in DMEM/F12 (1:1) medium supplemented with 10% fetal bovine serum (FBS), two mM L-glutamine, 100 IU/mL penicillin, and 100 µg/mL streptomycin under a humidified atmosphere containing 5% CO2 at 37 degrees Celsius.

Following the attachment and growth of GCs with about 70–80% confluence, they were supplemented with fresh media (FBS free) having various concentrations (10^−12^, 10^−10^, 10^−8^ and 10^−6^ M) of NMS for 48 h or at the concentration of 10^−12^ for different time points (3, 6, 12, 24 and 48 h) or 10^−12^ M singly for 24 h. For transfection, the NMUR1 siRNA-974, siRNA-774 (siNMUR1), siRNA-1100; the NMUR2 siRNA-650, siRNA-844 (siNMUR2), siRNA-1301, the YAP1 siRNA-1817 (siYAP1) [60] and the negative control siRNA (siControl) were purchased from Genepharma (Shanghai, China). The specific sequences are listed in Appendix A. The YAP1 overexpression plasmid (oeYAP1) and the pcDNA3.1 expression plasmid (oeContorl) were synthesized from Tsingke (Beijing, China). Both the siRNA and plasmid transfections into GCs were carried out with Lipofectamine™ 3000 transfection reagent (L3000150, Invitrogen, CA, USA) according to the manufacturer’s instructions. At least three replicates were tested for each experimental condition. The culture media were accumulated after 48 h and reserved for estrogen assay, and RNA and protein were extracted from cells.

### 4.4. Estrogen Secretion Determination

According to the manufacturer’s instructions, estrogen concentration in the collected culture media was measured by using Goat E ELISA KIT (Kamels, Shanghai, China). The sensitivity of the assays was 1.0 pg/mL, and the intra- and inter-assay coefficients of variation were less than 15%, respectively.

### 4.5. Quantitative real-Time PCR Analysis

Quantitative real-time PCR (qPCR) was conducted on an ABI 7500 Real-Time PCR System (Applied Biosystems, Carlsbad, CA, USA). Reactions were performed by using ChamQ™ Universal SYBR^®^ qPCR Master mix (Q711-02, Vazyme, Nanjing, China) and guaranteeing the total reaction system of 20 μL for each sample. The melting curve analysis was performed to verify amplification specificity. Primer sequences used to amplify the target genes are presented in Appendix A. Each experiment was independently repeated at least three times, and the fold change in the expression of each gene was quantified using the 2^−△△Ct^ method and normalized to glyceraldehyde-three-phosphate dehydrogenase (*GAPDH*) mRNA expression.

### 4.6. Western Blot Assay

Total protein was extracted from the GCs with RIPA lysis buffer and quantified via the BCA method (No. P0013B, and No. P0010, respectively, Beyotime, Shanghai, China). Total protein (20 μg/replicate) was separated on 12% SDS-polyacrylamide gel and electro-transferred to PVDF membranes (88520, Millipore, Bedford, MA, USA). After blocking with 5% non-fat dried milk, the membrane was incubated with the primary antibody overnight at four degrees Celsius and then incubated with the secondary antibody for one hour. Details of antibodies for this study are provided in Appendix A. After washing, the immunoreactive bands were detected with the Western Bright ECL kit (No. BL520B-2, biosharp, Guangzhou, China), and intensities were quantified by using Image Quant LAS 400 (Fijifilm, Tokyo, Japan). Each experiment was independently repeated at least three times, and target protein quantification was performed using Image J software (Wayne Rasband, MD, USA) with GAPDH as the internal control for normalization.

### 4.7. Immunohistochemistry Assay

The IHC assay was performed as previously described [61]. Briefly, ovarian paraffin sections (six mm thickness) were rehydrated through graded ethanol series. The tissue antigen was activated in citrate-buffered solution, and endogenous peroxidase was quenched in methanol with 3% hydrogen peroxide for 10 min after permeabilizing with 0.3% Triton X-100. The sections were blocked in 5% bovine serum albumin (BSA) for one hour and incubated with the primary antibody at four degrees Celsius overnight. The secondary antibody and third antibody at room temperature, according to the manual of the SABC-AP (rabbit IgG) Kit (SA1052, Boster, Wuhan, China) were added. After Diaminobenzidine (DAB) staining according to the DAB horseradish peroxidase color development kit (DA1015, Solarbio, Beijing, China), the samples were counterstained with Mayer’s hematoxylin (G1080, Solarbio, Beijing, China) for approximately 30 s, dehydrated and dried, then sealed with neutral resin. Positive signals appeared brown, and the images were acquired using a Nikon microscope (T300, Nikon, Tokyo, Japan).

### 4.8. Immunofluorescence Assay

Immunofluorescence (IF) was conducted as previously described [62]. In short, GCs were fixed in 4% paraformaldehyde (No. P0099, Beyotime, Shanghai, China) for 30 min at room temperature, then permeabilized with 0.5% Triton X-100, followed by blocking in 5% BSA. After incubation with the primary antibody overnight at four degrees Celsius for 14 h, the secondary anti-rabbit antibodies were incubated for two hours at room temperature. Lastly, cells were incubated with four’, six-diamidino-two-phenylindole (DAPI, No. C1006, Beyotime, Nantong, China) and mounted with coverslips. Negative controls were incubated with normal fetal bovine serum instead of the primary antibody. Positive signals appeared green, and the images were taken using the confocal laser scanning microscopy (Zeiss, Oberkochen, Germany).

### 4.9. Fluorescent Microscopy

Cellular mitochondria were fluorescently labeled through Mito-Tracker Red CMX-Ros (CS326, ZFdows, Nanjing, China). GCs were grown on a cell culture dish, and after 24 h of transfection and NMS treatment, cells were incubated with 250 nM of CMX-Ros and 1ug/mL Hoechst 33342 for 20 min at 37 degrees Celsius according to instructions. Images were observed via the confocal laser scanning microscope. The fusion versus fission mitochondria [63] and mitochondrial morphology [64] were quantified using standard parameters as previously described using ImageJ software (Wayne Rasband, MD, USA).

### 4.10. Transmission Electron Microscopy

GCs were suspended and fixed with a combination of 0.1 M sodium cacodylate, 4% paraformaldehyde, and 2% glutaraldehyde. Cell agarose blocks were dehydrated with a series of increasing concentrations of ethanol and then were infiltrated with resin and propylene oxide, embedded in resins, and polymerized overnight at 65 degrees Celsius. Samples were finally sectioned, stained, and examined under the transmission electron microscope. Quantification of cristae density (cristae number/mitochondrial area) with mitochondria from five cells or more was performed as previously described using ImageJ software [58].

### 4.11. Detection of Mitochondrial Membrane Potential

Cell mitochondrial membrane potential (MMP) was observed by using a Mitochondrial membrane potential detection kit (JC-1) (M8650, Solarbio, Beijing, China). GCs were washed and incubated with JC-1 working solutions for 30 min at 37 degrees Celsius in the incubator. Meanwhile, those treated with CCCP were used as a negative control during MMP evaluation according to the instructions. After incubation, live cells were imaged on a confocal laser scanning microscope. At high mitochondrial membrane potential, JC-1 forms JC-1 aggregates exhibiting red fluorescence, whereas the JC-1 monomer shows green fluorescence. Rapid-exposure confocal images were quantified to the mean fluorescence intensities of arbitrary regions through ImageJ (Wayne Rasband, MD, USA).

### 4.12. Measurement of Cellular Reactive Oxygen Species

For cellular reactive oxygen species (ROS) determination, the indicated GCs were seeded in 20 mm Petri dishes and incubated with the fluorescent probe DCFH-DA (S0033, Beyotime, Shanghai, China) and Hoechst 33342 at 37 degrees Celsius for 20 min according to the manufacturer’s instructions. The fluorescence intensity was detected using a confocal laser scanning microscope with excitation at 488 nm and emission at 525 nm.

### 4.13. Biochemical Analysis of Mitochondrial Respiratory Chain (MRC) Complexes

GCs were extracted and centrifuged, the precipitation was collected and ultrasonically broken via ice bath, and the activity of citrate synthase (CS), complex I (CI), com-plex II (CII), complex III (CIII), and complex IV (CIV) was measured on an ultraviolet–visible light spectrophotometer (UV-2450, SHIMADZU, Japan) using the mitochondrial complex object activity detection Kit (AKOP005U-1, AKOP006C-1, AKOP007C-1, AKOP008C-1, AKAC007C-2, boxbio, Beijing, China).

### 4.14. Quantification of Adenosine Triphosphate Content

The adenosine triphosphate (ATP) detection kit (S0026, Beyotime, Shanghai, China) was used to assess cellular ATP level. Briefly, after the GCs were lysed and centrifuged, the supernatant was collected and then mixed with ATP detection working diluent. Luminance value (RLU) was acquired through chemiluminescence meter (E5311, Promega BioSystem, Sunnyvale, CA, USA) to generate standard curve, and the ATP content in the samples was calculated from the standard curve.

### 4.15. Immunoprecipitation Assay

GCs were lysed in lysis buffer (20118ES60, Yeasen, Shanghai, China) and rocked gently at four degrees Celsius for 30 min. Following centrifugation at 16,000× *g* for 30 min, the supernatant was subjected to immunoprecipitation with Protein A/G Magnetic Beads (sc-2003, Solarbio, Shanghai, China) at four degrees Celsius for one hour after pre-clearing the beads. The tag antibody was incubated with the cell lysates with gentle rotation overnight at four degrees Celsius. The samples were then washed three times with lysis buffer. SDS sample buffer was added to the beads, which were boiled for five minutes, and the respective input lysates were analyzed by Western blot.

### 4.16. Statistical Analysis

All experiments were repeated at least three times. Statistical analyses were analyzed with SPSS 24.0 (SPSS Inc., Chicago, IL, USA) and data quantifications are presented as mean values ± S.E.M (standard error of mean). Comparisons between two independent groups were determined by t-test. Comparisons among multiple groups were determined by one-way ANOVA with Tukey’s test. Values of *p* < 0.05 are considered as statistically significant.

## 5. Conclusions

In conclusion, our current study informs that NMS maintains mitochondrial morphology and function through NMUR2 to regulate steroidogenesis in goat ovarian GCs, and the possible mechanism of these is managed by the YAP1–PPARGC1A pathway. Our information verifies that NMS is a crucial regulator to regulate the function of ovarian GCs, thereby providing the mechanism reference for follicular development and ovulation.

## Figures and Tables

**Figure 1 ijms-23-13402-f001:**
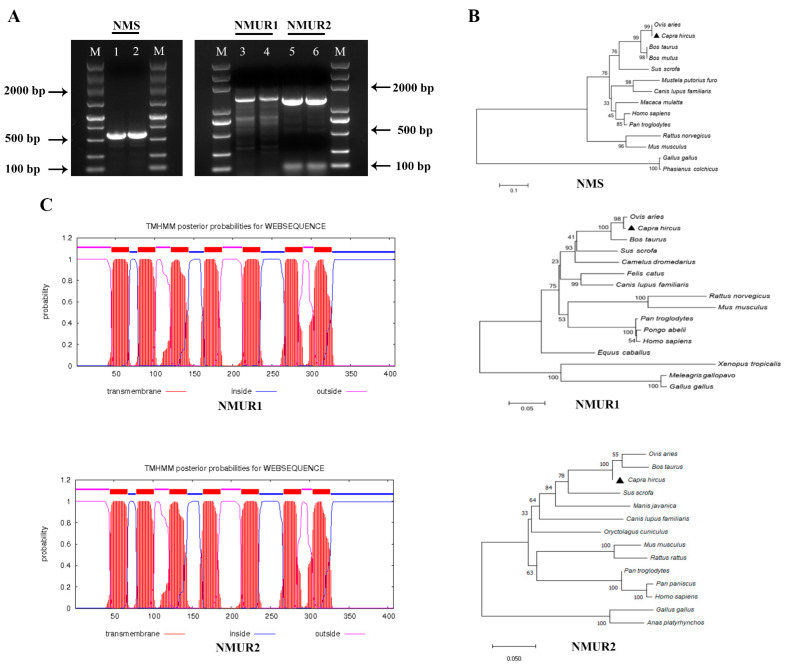
Cloning of NMS and its receptor and sequence analysis. (**A**) Gel electrophoresis of goat NMS and its receptor PCR products. Lane 1, 2: NMS; Lane 3, 4: NMUR1; C: Lane 5, 6: NMUR2; M: DNA Marker DL 5000. (**B**) Phylogenetic tree of the goat NMS and its receptor nucleotide sequences. (**C**) Transmembrane region of the goat NMUR1andNMUR2 amino acid sequence fragment.

**Figure 2 ijms-23-13402-f002:**
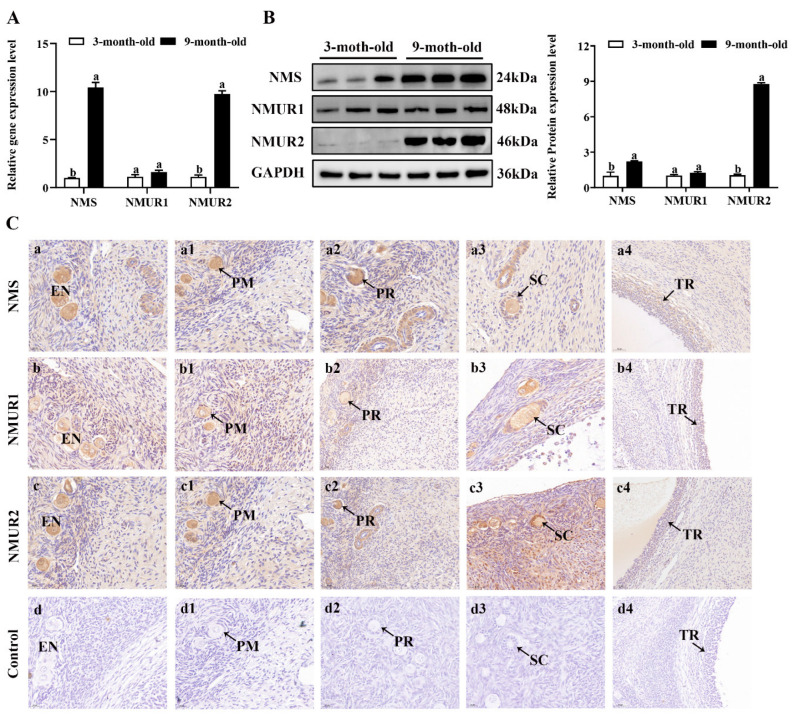
Localization and expression of NMS and its receptor in goat ovaries. (**A**) The mRNA expression of NMS and its receptor in three-month-old and nine-month-old goat ovaries were detected by qPCR. (**B**) The protein expression of NMS and its receptor in three-month-old and nine-month-old goat ovaries were detected by Western blot. (**C**) Localization of NMS and its receptor proteins in goat ovaries. Tissue sections were stained with specific antibodies and counterstained with hematoxylin. Positive signals appeared brown, and counterstaining background appeared blue. a–a4: stained with NMS polyclonal antibody; b–b4: stained with NMUR1 polyclonal antibody; c–c4: stained with NMUR2 polyclonal antibody; d–d4: negative control (no significant immunoreactivity was observed with normal fetal bovine serum instead of the primary antibody); EN: oocyte nest; PM: primordial follicle; PR: primary follicle; SC: secondary follicle; TR: tertiary follicle; scale bar 50 μm. (**D**) Localization of NMS and its receptor in cultured goat GCs. Blue color: DAPI-stained nuclei; red color: immunofluorescence representing the reaction of antibodies with antigens. (**E**) The mRNA expression of NMS and its receptor in different diameters of follicles were detected by qPCR. (**F**) The protein expression of NMS and its receptor in different diameters of follicles were detected by Western blot; data are expressed as fold change and are presented as mean ± S.E.M. of the results obtained in at least three independent experiments. Different letters indicate significant differences in the expressions between the groups (*p* < 0.05).

**Figure 3 ijms-23-13402-f003:**
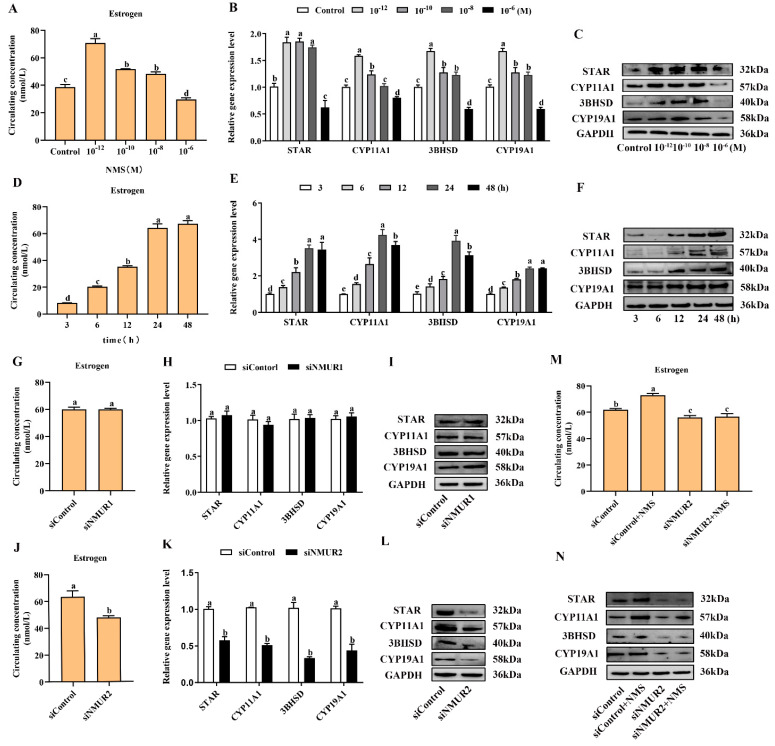
Effects of NMS treatment and knockdown of NMUR2 on steroid hormone synthesis in goat granulosa cells. (**A**) The level of estrogen secretion under NMS treatment at different concentrations (10^−12^, 10^−10^, 10^−8^ and 10^−6^ M) in goat GCs. (**B**) The mRNA expressions of STAR, CYP11A1, 3BHSD, and CYP19A1 under NMS treatment at different concentrations were detected by qPCR. (**C**) The protein expressions of STAR, CYP11A1, 3BHSD, and CYP19A1 under NMS treatment at different concentrations were detected by Western blot. (**D**) The level of estrogen secretion under NMS (10^−12^ M) treatment for different treatment times (3, 6, 12, 24 and 48 h) in goat GCs. (**E**) The mRNA expressions of STAR, CYP11A1, 3BHSD, and CYP19A1 under NMS (10^−12^ M) treatment at different treatment times were detected by qPCR. (**F**) The protein expression of STAR, CYP11A1, 3BHSD, and CYP19A1 under NMS (10^−12^ M) treatment at different treatment times by Western blot. (**G**) The effect of siNMUR1 treatment on estrogen secretion level. (**H**) The effect of siNMUR1 treatment on STAR, CYP11A1, 3BHSD, and CYP19A1 mRNA expression. (**I**) The effect of siNMUR1 treatment on STAR, CYP11A1, 3BHSD, and CYP19A1 protein expression. (**J**) The effect of siNMUR2 treatment on estrogen secretion level. (**K**) The effect of siNMUR2 treatment on STAR, CYP11A1, 3BHSD and CYP19A1 mRNA expression. (**L**) The effect of siNMUR2 treatment on STAR, CYP11A1, 3BHSD and CYP19A1 protein expression. (**M**) The effect of siNMUR2 and NMS cotreatment on estrogen secretion. (**N**) The effect of siNMUR2 and NMS cotreatment on STAR, CYP11A1, 3BHSD, and CYP19A1 protein expression. Data are expressed as fold change and are presented as mean ± S.E.M. of the results obtained in at least three independent experiments. Different letters indicate significant differences in the expressions between the groups (*p* < 0.05).

**Figure 4 ijms-23-13402-f004:**
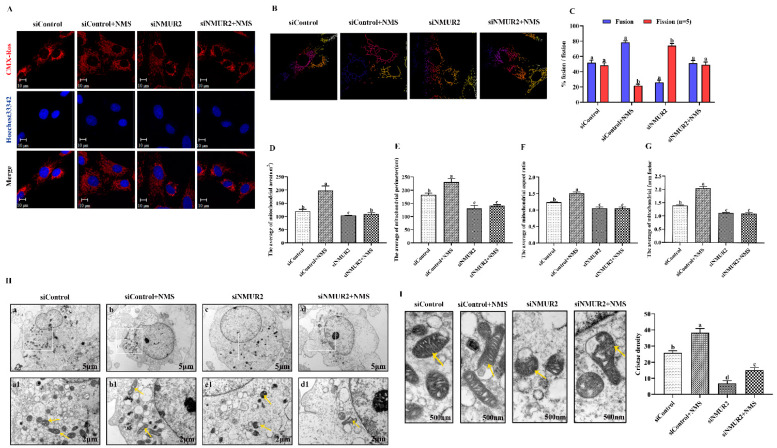
NMS affects mitochondrial morphology and genes related to mitochondrial quality in goat granulosa cells. (**A**) Confocal scanning image micrographs of mitochondria in goat GCs under NMS with or without siNMUR2 treatment. Mitochondria stained (red) with CMX-Ros, and nucleus (blue) stained with Hoechst 33342. (**B**,**C**) Statistical representation of fusion and fission mitochondria analyzed by the software. Analysis of mitochondria area (**D**), mitochondrial perimeter (**E**), mitochondrial aspect ratio (**F**), mitochondrial form factor; mitochondrial form factor is calculated as (perimeter2)/(4π⋅surface area) and reflects the complexity and branching aspect of mitochondria (**G**). (**H**) Representative electron micrographs of mitochondria in goat GCs under NMS with or without siNMUR2 treatment; a1–d1 are enlarged images of a–d; arrowheads indicate mitochondria. (**I**) Electron micrographs of the mitochondrial cristae and quantification of cristae density in goat GCs under NMS with or without siNMUR2 treatment. Arrowheads indicate mitochondrial cristae changes. The number of cristae was counted for each mitochondrion and divided by the mitochondrial area in megapixels. Four to twelve mitochondria were analyzed in each cell. n = 32, 31, 29, and 27 in siControl, NMS, siNMUR2 and NMS + siNMUR2 groups, respectively. (**J**) The mRNA expression of genes related to mitochondrial dynamics was determined by qPCR. (**K**) The protein expression of genes related to mitochondrial dynamics was determined by Western blot. Data are representative of at least three independent experiments. Different letters indicate significant differences in the expressions between the groups (*p* < 0.05).

**Figure 5 ijms-23-13402-f005:**
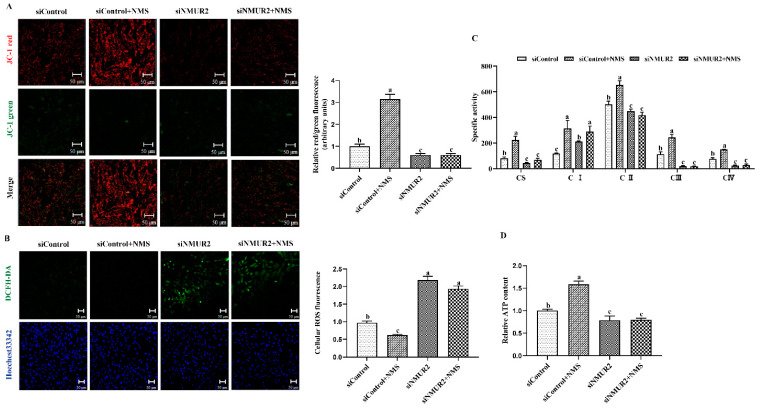
NMS regulates mitochondrial function via mitochondrial unfolded protein response in goat granulosa cells. (**A**) Representative images (left) and statistical quantification (right) of the mitochondrial membrane potential results. The goat GCs were stained with JC-1 under NMS with or without siNMUR2 treatment. (**B**) Representative images (**left**) and statistical quantification (**right**) of ROS production assay results. The goat GCs were stained with DCFH-DA under NMS with or without siNMUR2 treatment and observed under a fluorescence microscope. DCFH-DA: green; Hoechst 33342: blue. (**C**) MRC activities in goat GCs under NMS with or without siNMUR2 treatment are expressed as nmoles/min/mg of protein. CS, citrate synthase; CI–IV, MRC complexes I–IV. (**D**) Intracellular ATP level after one hour in goat GCs under NMS with or without siNMUR2 treatment. (**E**) The mRNA expression of genes related to mitochondrial unfolded protein response marker genes was determined by qPCR. (**F**) The protein expression of genes related to mitochondrial unfolded protein response marker proteins was determined by Western blot. Data are representative of at least three independent experiments. Different letters indicate significant differences in the expressions between the groups (*p* < 0.05).

**Figure 6 ijms-23-13402-f006:**
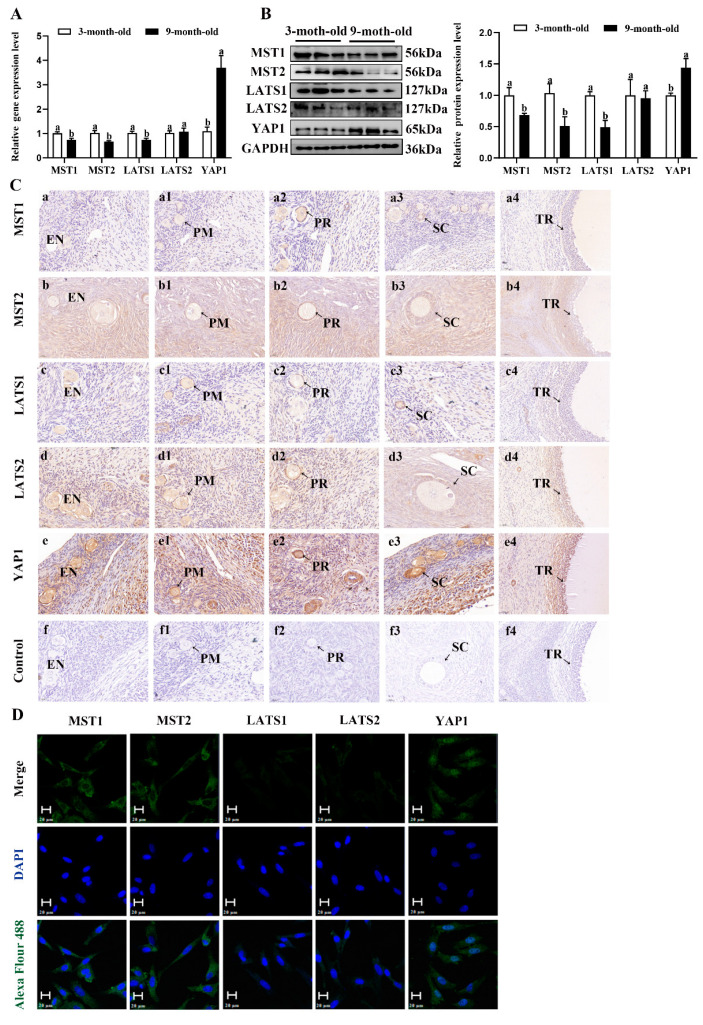
Localization and expression of the Hippo pathway core components in goat granulosa cells. (**A**) The mRNA expression of the Hippo pathway core components (MST1, MST2, LATS1, LATS2, and YAP1) in three-month-old and nine-month-old goat ovaries was detected by qPCR. (**B**) The mRNA expression of the Hippo pathway core components in three-month-old and nine-month-old goat ovaries was detected by Western blot. (**C**) Localization of the Hippo pathway core components proteins in goat ovaries. Positive signals appeared brown and counterstaining background appeared blue. (**a**–**a4**): stained with MST1 polyclonal antibody; (**b**–**b4**): stained with MST2 polyclonal antibody; (**c**–**c4**): stained with LAST1 polyclonal antibody; (**d**–**d4**): stained with LATS2 polyclonal antibody; (**e**–**e4**): stained with YAP1 polyclonal antibody; (**f**–**f4**): negative control (no significant immunoreactivity was observed with normal fetal bovine serum instead of the primary antibody); EN: oocyte nest; PM: primordial follicle; PR: primary follicle; SC: secondary follicle; TR: tertiary follicle; scale bar 50 μm. (**D**) Localization of the Hippo pathway core components in cultured goat GCs. Blue color: DAPI-stained nuclei; green color: immunofluorescence representing the reaction of antibodies with antigens; (**E**) The mRNA expression of the Hippo pathway core components in different diameters of follicles was detected by qPCR. (**F**) The protein expression of the Hippo pathway core components in different diameters of follicles was detected by Western blot; the data are presented as mean ± S.E.M. value from at least three individuals. Different letters indicate significant differences in the expressions between the groups (*p* < 0.05).

**Figure 7 ijms-23-13402-f007:**
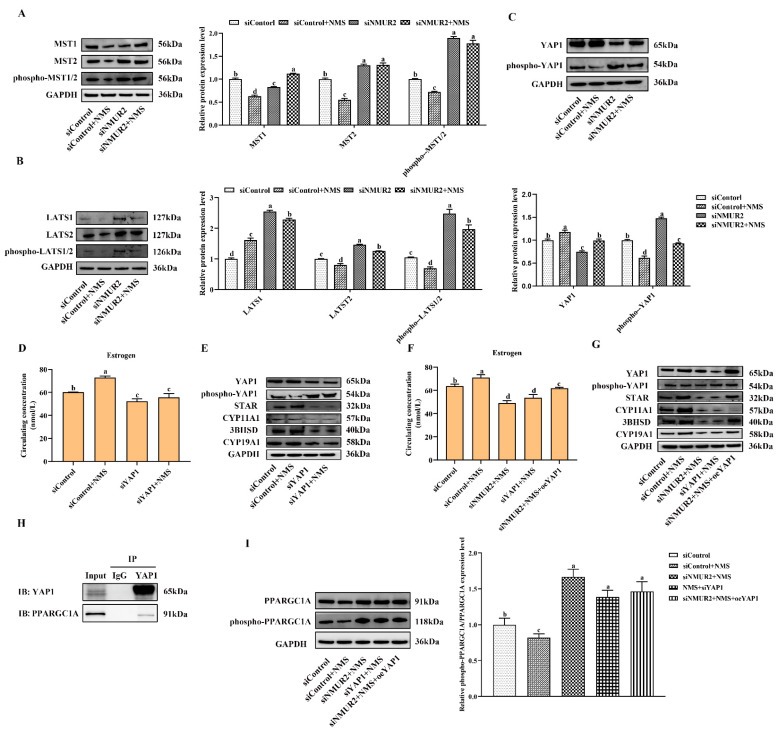
NMS binds to NMUR2 to promote steroidogenesis via Hippo pathway in goat granulosa cells. (**A**–**C**) The expression of MST1, MST2, phospho-MST1/2(Thr183), LATS1, LATS2, phospho-LATS1/2(Ser909/872), YAP1, and phospho-YAP1(Ser127) in goat GCs under NMS with or without siNMUR2 treatment were determined by qPCR. (**D**) The level of estrogen secretion under siYAP1 treatment in goat GCs. (**E**) The protein expression of STAR, CYP11A1, 3BHSD and CYP19A1 under siYAP1 treatment in goat GCs was determined by Western blot. (**F**) The level of estrogen secretion under siNMUR2 and oeYAP1 cotreatment in goat GCs. (**G**) The protein expression of STAR, CYP11A1, 3BHSD and CYP19A1 under siNMUR2 and oeYAP1 cotreatment was determined by Western blot. (**H**) Coimmunoprecipitation analysis of YAP1 protein with PPARGC1A protein. (**I**) The expression of phospho-PPARGC1A(Ser571)/PPARGC1A. Data are expressed as fold change and are presented as mean ± S.E.M. value from at least three individuals. Different letters indicate significant differences in the expressions between the groups (*p* < 0.05).

**Figure 8 ijms-23-13402-f008:**
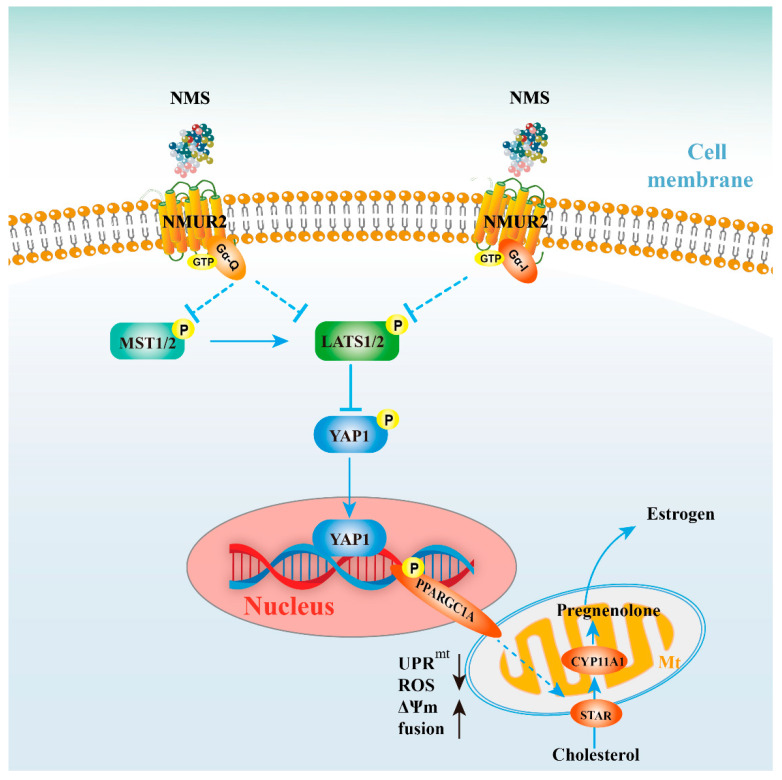
Schematic diagram for NMS regulates steroidogenesis, possibly via the NMUR2/YAP1/PPARGC1A pathway, in ovarian granulosa cells. NMS in GCs regulates steroidogenesis by affecting mitochondrial morphology, mitochondrial unfolded protein response, mitochondrial membrane potential, the activities of mitochondrial respiratory chain complexes, adenosine triphosphate level, and cellular reactive oxygen species production, which is possibly mediated through the NMUR2/YAP1/PPARGC1A pathway.

## Data Availability

The study did not report any data.

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
