# Peer review of "Neuromedin S Regulates Steroidogenesis through Maintaining Mitochondrial Morphology and Function via NMUR2 in Goat Ovarian Granulosa Cells"

_ijms, 2022, doi:10.3390/ijms232113402_

Round 1
Reviewer 1 Report
In their submitted manuscript Sun et al. Addressed the possible role of neuromedin S (NMS) on goat ovarian granulosa cells steroidogenesis and mitochondrial functions. They also proposed the possible mechanisms of NMS action in ovary via signaling/crosstalk with the elements of Hippo pathway.
I think this is an interesting manuscript with scientific merit and individual experiments seems to be well conducted and well supported by attached primary data archive. On the other hand, I consider some crucial parts of the text lack well elaborated rationale and the mechanistic explanation of the NMS action via Hippo pathway signaling/crosstalk lacking robust experimental evidence and suggest re-considering these parts before the decision about the publication can be made.
Major points:
1) Since the authors aim to probe new areas of experimental (ovarian) physiology I would expect that classical physiological model systems (mice, rat) will be utilized instead of more rare goat model and/or those model will be added to the experiment for potential comparison. Furthermore, since authors performed experiments on cell cultures, the human cell ovarian cell lines can be utilized. The absence of those model systems strongly decrease the overall impact of the study in areas, where authors obtained interesting and relatively robust experimental evidence (effect on steroidogenesis and mitochondrial structure/function).
2) A strong the rationale of the author´s focus on Hippo signaling pathway have to be provided. Why they decided to focus on Hippo pathway? Did authors probe also other signaling pathways?
3) Also the experimental evidence connecting the NMS and Hippo signaling pathway is weak and highly speculative. While in the text authors use appropriate language (could be etc.), they statement from the title “function via NMUR2/YAP1/PPARGC1A pathway” should be removed since the current experimental evidence (different protein levels from (3?) experiments one immunoprecipitation experiment 5H) do not constitute enough experimental evidence. Since the pilot bioinformatic analysis do not suggest the crosstalk of NMS/Hippo pathway by factor 10 (lower) establishment of this concept will require much more robust approaches on model systems, since it is not expected to be restricted to goat granulosa cells.
I suggest here to include the results addressing the effect of NMS on steroidogenesis and mitochondria in Results part and parts addressing mechanisms/Hippo pathway at the end of Results as “possible mechanisms explanation” or more preferentially to Discussion part (with corresponding figures as Supplements).
4) Finally, if authors believe they will be able to experimentally establish the connection between NMS/Hippo signaling I strongly encourage them to study them further on model systems (human (probably neural) cell lines and with robust experimental (e.g. protein-protein interaction) design.

Reviewer 2 Report
Overall, the article is well structured and the results obtained present a novel advance from the point of view of the ovaries given the paucity of research on Neuromedin S in this organ.
I have observed some elements that, in my opinion, should be revised for a better understanding of the text.
In the introduction.
In lines 51 to 53 it is written "The results obtain consistently report that NMS expression is mainly observed in the brain, NMUR1 mainly presents in the periphery, and NMUR2 mainly occurs in the central nervous system". When you say "periphery" do you mean the periphery of the brain or the peripheral nervous system? This needs to be clarified or rephrased to make it more understandable.
In the "Results" section.
In general, the figures are too small to be easily read. I don't know if this is due to the editing by the journal or if they have been sent with that size. If it is the second case, they should have a larger size to be more understandable.
In Figure 2, C) is so small that it is not possible to clearly distinguish the positivity of the cells by IHC. In addition, the meaning of the acronyms in the images is not specified. In the caption of figure 2 F) (fluorescence) it should be specified that the cells are in culture and are not pictures of an ovarian section or ovarian follicles as in C. On the other hand, I would change the order of the text so that the image F is after C and then change F to D. This way the whole microscopy part would be followed and it is easier to follow in the text and in the figure.
Figure 4 is exactly the same as figure 2, so I would advise the same changes. In addition, in image B, the LATS1 bands in the 3-month-old animals are not clear and it is not possible to clarify whether the difference in the relative expression level between 3 and 9 months is true or whether it is a product of the "stain" observed in the Western blot. In E) something similar occurs in LATS1. I believe that these Western blots should be repeated so that the bands are cleaner and do not generate doubts.
In figure 6 (B), the size of the photos should be larger to be able to see in detail the arrangement of the mitochondria in the cells.
In figure 6 H and I) the same thing happens, although the electron microscopy images seem to be good, due to their small size it is very difficult to see what is really important.
In Figure 7 A and B) there is the same problem or worse. The pictures are so small that having them and not having them would be the same because nothing is visible. Since a relative quantification of the fluorescence has been done, the images should look good.
In section 2.4, line 205 the follicles of different diameter have always been selected from the 9-month-old animals, right? It would be good to specify this so that there is no doubt about it.
In line 244 it is written Fig.3M-O when in figure 3 there is no image O. Please correct it. Also, there seems to be no concordance with the reference of the figures in the text and in the figure itself. Please check it.
In Material and methods:
Was the selection of ovarian follicles of different sizes performed exclusively in the 9-month-old animals or also in the 3-month-old animals? This should be specified so as not to create doubts.
In Abbreviations:
Mitofusin 1 has no abbreviation while Mitofusin 2 is as OPA1. Is that correct?
We also find MFN 1 and MFN 2 with no associated name.
Please, if a protein/gene has no abbreviation remove it from the list and every abbreviation should have its name.
Otherwise, I consider it a very complete work and it opens up other questions that I am sure will be answered by your group in future publications.
Round 2
Reviewer 1 Report
Authors have addressed my reservation and some remaining aspects will be further addressed in their future studies.